# Development of a High-Quality ELISA Method for Dinotefuran Based on a Novel and Newly-Designed Antigen

**DOI:** 10.3390/molecules24132426

**Published:** 2019-07-02

**Authors:** Fangfang Zhao, Jingkun Liu, Jinhui Luo

**Affiliations:** 1Analysis & Testing Center, Chinese Academy of Tropical Agricultural Sciences, Haikou 571101, Hainan, China; 2Laboratory of Quality & Safety Risk Assessment for Tropical Products(Haikou), Ministry of Agriculture, Haikou 571101, Hainan, China; 3Hainan Provincial Key Laboratory of Quality and Safety for Tropical Fruits and Vegetables, Haikou 571101, Hainan, China; 4Quality Supervision and Inspection Center of Tropical Agro-Products, Haikou 571101, Hainan, China; 5Institute of Environment and Plant Protection, Chinese Academy of Tropical Agricultural Sciences, Haikou 571101, Hainan, China

**Keywords:** dinotefuran, Hapten design, spatial configurations, ELISA, sensitivity and specificity

## Abstract

The structure of hapten determines the performance of the antibody and the corresponding detection method. A new type of antigen was designed and synthesized to expose the spatial and characteristic structure of dinotefuran molecule, and a type of high-quality antibody was obtained. The IC_50_ value of the monoclonal antibody was 5.30 ng/mL and its cross-reactivity (CRs) was less than 2% when reacting with other structurally related analytes. The effects of spatial configurations of hapten on the antibody were visually analyzed while using the appropriate software according to the quality of the antibodies, which showed that the specificity of the antibody is closely related with the exposed structure of hapten. An ELISA assay with an IC_50_ of 5.66 ng/mL and a linear range of 1.95 to 16.29 ng/mL was developed. The results that were obtained from the ELISA and HPLC methods were equivalent. The results showed that spatial simulation is a crucial method that is used in the designing of hapten to obtain a sensitive and specific antibody. The application of this method will highlight the potential aim and improve the detection efficiency of ELISA.

## 1. Introduction

Dinotefuran is a class of pesticide that belongs to the third generation of neonicotinoids [1,2,3]. Its molecular structure is significantly different from the first and second generations, which include imidacloprid and thiamethoxam. Chloropyridine or chlorothiazole is a common constituent of neonicotinoids, which is replaced by the tetrahydrofuran group in dinotefuran. In addition, dinotefuran does not contain halogen atoms in its molecular structure, unlike the other two generations of neonicotinoids. The use of dinotefuran increased in many agricultural practices due to its broad insecticidal spectrum since imidacloprid, thiamethoxam, and clothianidin were banned in the European Union (EU) for their damaging effects on bee populations [1,4,5]. Although dinotefuran rapidly degrades, its prolonged use often leaves some residues in foods. Therefore, chronic consuption of contaminated food poses a threat to human health [6,7].

The current analytical methods [8] that are used to measure dinotefuran residues in agricultural products or in the environment are: gas chromatography (GC), GC-Mass spectrometry (GC-MS), liquid chromatography (LC), LC-MS, and enzyme-linked immune assay (ELISA). Although these instrumental methods have high precision, accuracy, and sensitivity, some drawbacks, such as high costs and complex operational procedures, limits their application [9,10]. Moreover, these methods are time-consuming and not suitable for agricultural products, especially fruits and vegetables, which have a short shelf-life. However, ELISA, which is the most widely used rapid detection method, overcomes these problems, especially in screening and field detection.

As the basis of immune responses, antigen and antibody determine the sensitivity and specificity of the ELISA methods. The body produces antibodies in response to an antigen and they are specific to the antigen that causes the immune response. Therefore, the design of an antigen is the most critical step in the development of ELISA methods. The process of designing an antigen involves the simulation of many analytes to ensure that the synthesized antigen contains the required characteristics or structures of the analytes to effectively elicit an immune response. Thus, the binding sites for coupling and the length of the arms would determine whether the structures could be exposed to the body and could incur the immune response or not. The binding sites and the length of the arms should be preliminarily predicted and analyzed to ensure the specificity of the antibody-analyte binding, which is the foundation of the ELISA-based methods.

The molecular of dinotefuran mainly contains two characteristic structures, furan and guanidine. According to the reports, the two parts in other analytes could be used for the character structure to obtain the antibody. However, guanidine is shared with many types of anlytes, thus the specificity of the antibody would be lowered when it is used for character structure to obtain antibody. For example, an ELISA method [11] for detecting the dinotefuran residue based on the guanidine molecule was previously reported. This method introduced an antibody with high sensitivity and specificity to many analytes, except clothianidin. Dinotefuran and clothianidin are highly similar when the furan structure of dinotefuran was linked with protein and guanidine was performed as the reaction structure. Thus, the specificity of the antibody was extensively lowered. We designed a new type of hapten based on the characteristic group of the dinotefuran, furan nucleus, to design an antibody with high-specificity and to investigate the impact of the location of the binding site on the antibody. The results showed that a high-quality antibody for ELISA was obtained with high sensitivity and specificity to all structurally similar analytes, including clothianidin. The binding sites were analyzed in detail based on the spatial structure of the analytes and the detection results. This study not only supplied high-quality antibody for the detection of dinotefuran, but it also provides the theoretical knowledge for the selection and design of an antigen.

## 2. Results and Discussion

### 2.1. Design and Synthesis of Haptens and Antigens

Dinotefuran is a new class of neonicotinoid pesticides and it has two major parts in structure, furan and guanidyl. It is the only pesticide that contains furan and no halogen atom. A previous study designed an antigen by linking clothianidin with a protein by adding an arm to the thiazole ring. In the report, the antigen outstood the guanidyl group of thehapten, and the antibody showed high sensitivity and specificity, but unspecific affinity to dinotefuran and clothianidin. Thus, in this work, the furan group was exposed to the animal body via linking the guanidyl group to the protein. Based on the spatial effects of the protein on the antigen, a C_4_ chain was introduced to serve as the arm. The hapten and antigen were synthesized while using routine methods and were characterised by the NMR techniques (see Appendix A) before further processing.

Hapten mimics the analytes and, therefore, stimulates the body to produce antibodies. The high similarity between hapten and analytes would cause the production of high affinity antibody to the analytes. The characteristic group, spatial structure, and the similarity to other analytes are important factors that affect the quality of the antibody [12]. Thus, the binding site and the arm that links hapten with the carrier protein determined the spatial structure of the hapten, which is a crucial determinant of whether the body would react with hapten or not. Therefore, the selection or introduction of a proper binding site and arm that links hapten to the carrier protein is the first step in this work. Further multianalysis indicates that hapten should accurately reflect the common analytes to ensure increased performance [13]. In addition, the sensitivity and cross-reactivity should be considered during the design of antigen. Exposing the characteristic structure of analyte or hapten to the body could improve the quality of the antibody. In this course, using software to compare spatial structures of potential hapten to that of analyte is an effective way of obtaining satisfied hapten. The main structure of dinotefuran was introduced to the body in form of the antigen designed in this paper to obtain a high specific antibody against dinotefuran. Hapten reflected the main parts and the spatial structure of dinotefuran, according to the software analysis.

### 2.2. Positive Monoclonal Hybridomas from the Semisolid Medium Screening

A semisolid medium was used for screening cell fusion because of the many disadvantages of the liquid medium in cloning of hybridomas. For example, in a liquid medium, the rapid-growing cells, including the nonproducing cells, dominate and spread, covering the slow-growing cells, including the antibody-producing clones in the mixture. This limits the yield of the desired hybridomas.

The clones are initially monoclonal, eliminating the need for limited dilution, as described by Davis [14]. Using this method, the clonal cells require a shorter time to produce Mc-Abs, reaching up to 28 days, and the time that is taken to perform the technical work is reduced by approximately 33%. The coating antigen (2 μg/mL of) and the 10 folds dilution of the clone supernatant were used in the non-competitive indirect ELISA and CI-ELISA assays. Finally, three monoclonal hybridomas, 2D6, 2F10, and 3C7, which produce antibodies against dinotefuran, were obtained, with the corresponding OD_450_ values of 0.65 ± 0.08, 0.77 ± 0.06, and 1.05 ± 0.03, respectively.

### 2.3. Affinity, Sensitivity, and Specificity of the Antibody

The affinity, sensitivity, and specificity, as the indicators of the performance of the antibody or cell, were titer [15,16], IC_50_, and CRs, respectively. These values are shown in Table 1 and Table 2. The titer was determined while using the checkboard method when the coating antigen was 2 μg/mL and the OD_450_ was approximately 1 (0.95–1.05). This step provided the concentrations of the antibody and antigen that were used to determine the IC_50_ and calculate the CR. The highest affinity and sensitivity were recorded in the clone 3C7, as shown in Table 1. The values of this clone 3C7 were approximately 4–5 times higher than the other two clones, particularly with regard to its sensitivity, which was the most critical factor in the assay methods.

Table 2 shows the specificities of the antibodies, as represented by the CR in the CI-ELISA system. Acetamiprid, nitenpyram, diuron, thiamethoxam, tiotropium, and tetrahydrofuran were used as the standards to test the specificity of the antibody, according to the molecular structure of the dinotefuran. Although all of the antibodies were highly specific to most of the tested analytes, clone 3C7 showed the best performance, having the CRs of less than 2%, particularly the ones in which the halogen atoms were far from the protein.

### 2.4. The Analysis of the Coupling Site of the Hapten

As described above, guanidyl and furan are the key structural elements of dinotefuran. These elements are responsible for inducing an immune response when exposed to the body. When compared with the carrier protein and its spatial structure, the smallness of the antigen makes the portion of the antigen linked to the protein that is to be fully covered by the protein, and hence not recognizable by the body. Thus, the coupling sites are critical during the design of the antigen.

In a previous study [11], an arm was added to link a thiazole group with the protein, and the thiazole replaced the furan cycle to synthesize the hapten (Figure 1, hapten) to stimulate the body to produce antibodies. Based on the results, the antibodies with CRs of less than 0.1% were produced, which showed higher specificity than other neonicotinoid insecticides. However, the results showed that the CRs of the antibodies that were produced against clothianidin and dinotefuran were 64% and 184%, respectively. According to the structures, the results revealed that the antibodies only recognized the guanidyl groups that belonged to clothianidin and dinotefuran. Obviously, the low specificity of the antibody is closely related with the common guanidyl group in the molecular of the analyte. On the contrary, the main structure of dinotefuran, a furan group, was exposed in the designed antigen via guanidyl in this work. Antibodies with high specificity and sensitivity were obtained and, as we expected, the CR of dinotefuran and clothianidin were significantly low (<0.2%). The results showed that furan elicited an immune response from the body and it was the essential part of the molecular recognition process. Therefore, furan was used to confirm the specificity of binding between the antibody and antigen. As expected, the CR of the furan and dinotefuran was remarkably higher than the other compounds (Table 1). The results predicted that furan is an effective structure in the immune. At the same time, it can be speculated that, as a smaller structure than other analytes, the tetrahydrofuran is more easily identified and docked by the antibody for less spatial resistance.

The spatial structures of the analytes were analyzed to further explore the impact of the coupling sites on the antigen and sensitivity of the antibodies, as shown in Figure 2. Depicted in column A, the guanidyl of dinotefuran and clothianidin showed similar spatial configurations at different perspectives when their heterocycles were linked to the protein, as described in the report. In contrast, the spatial configurations of their heterocycles were significantly different (red circle in column B of Figure 2) from all perspectives when they were linked to the protein. The results of this study and the three-dimensional (3D) view of the molecular structure are consistent with each other. Based on dinotefuran and nitenpyram, as depicted in column C of Figure 3, the 3D views of the heterocycles were significantly different when the guanidyl was the coupling site although the molecules shared quite similar structures. According to the specificity of the antibodies in previous report and this work, it occurs to us that the spatial structure of the hapten affects the identification of the body to the analyte. A similar structure of clothianidin and dinotefuran led to the low specificity of the antibody in a previous report, while a quite opposite result in this work.

Notably, the common structure—guanidyl group–of clothianidin and dinotefuran exerted nearly no effect on the specificity of the antibody in this work. It can be speculated that the protein enclosed the guanidyl group and it could not induce an immune response. On the contrary, exposing the guanidyl residue to the body induced antibody production when the cycle group was linked to the carrier protein. Therefore, although guanidyl is present in many of the neonicotinoids, it can be deduced that the high affinity of the furan cycle may have greatly improved the specificity of the antibody, while the guanidyl provided a suitable coupling site for the protein.

### 2.5. Establishment of an Optimized CI-ELISA

Although it showed the best performance among the antibodies, the conditions of the CI-ELISA for clone 3C7 were further optimized. For solvent optimization, various concentrations of methanol, which is often used as co-solvent in immunoassays to improve the solubility of analytes, were tested to determine the best sensitivity. The lowest IC_50_ (5.83 ng/mL) was obtained at 20% methanol, as indicated in Figure 3A. We speculate that low concentrations of methanol affected the solubility of the analytes, whereas high proportions of methanol reduced the activity of the antibody.

When compared with PBST, PBS was identified as superior reaction buffer (Figure 3B) and, finally, a lower IC_50_ (5.30 ng/mL) was obtained, which showed that the tween exerted a negative influence on the reaction. Through these optimizations, a typical standard curve for the dinotefuran immunoassay was established, as shown in Figure 3C. The average IC_50_ for dinotefuran was 5.30 ng/mL and its linear range was from 1.95 to 16.29 ng/mL.

### 2.6. Analysis of Spiked Samples

A spiked recovery experiment was performed to evaluate the accuracy of the developed assay method. Cabbages from a local supermarket were spiked with three concentrations of dinotefuran (1, 5, 10 ng/g) and then pre-treated using the QUENCHERS method [17,18]. The dinotefuran recovery shown in Table 3 ranged from 87% to 98%, which were highly suitable for determining the pesticide residues, although the RSDs were slightly higher than those of the HPLC method. The developed method showed high sensitivity, specificity, and accuracy, and it was suitable for the detection of trace levels of dinotefuran in some actual samples. In addition, the designed ELISA method can be further modified and used as a rapid detection technique in field testing.

## 3. Conclusions

When compared with the hapten in previous report, a new type of hapten of dinotefuran that highlighted the furan part as the character structure was designed and synthesized by linking it to the protein while using a C_4_ bridge. Immunizing mice with the new antigen triggered the production of antibodies with high sensitivity and specificity. Monoclonal antibodies were prepared from on the polyclone antibodies produced, which showed much higher quality, sensitivity, and specificity. Differently, the antibody that was obtained by eliciting the body while using the hapten linked with protein from the furan terminal showed low specificity to clothianidin, because the analytes shared highly similar structure when guanidine was exposed as characteristic structure to the body. According to these research evidences, the spatial structures of the antigen and the structurally similar analytes were analyzed to investigate the basis of the high quality of these antibodies. The results indicated that the quality of antibody was directly dependent on the spatial structure of the antigen that was exposed to the body. The spatial structure of the analyte was the basis of the quality of the antibody. These findings present a new perspective for designing high-quality antigens. Finally, an ELISA analytical method that was based on the antibody was developed, with an equivalent performance to the HPLC method.

## 4. Methods and Materials

### 4.1. General Experimental Reagents and Instruments

The standards of acetamiprid, dinotefuran, nitenpyram, diuron, and tetrahydrofuran were purchased from Alading (Shanghai, China). The methanol used for the HPLC was of a chromatographic grade, and other chemicals were obtained from Guoyao Co. Ltd. (Shanghai, China) were of the analytical grade, including phosphate buffered saline(PBS) and PBST. The main components of PBS are Na_2_HPO_4_, KH_2_PO_4_, NaCl and KCl. PBST means PBS with tween-20.

Thin layer chromatography (TLC) was performed on the F_254_ silica gel (100–200 mesh) plates and the results were detected by the ultraviolet (UV) light. The TLC silica gel plate and silica gel (for compounds purification) were obtained from Haiyang chemicals plant (Qingdao, China). Bovine serum bovine albumin (BSA), cell culture media RPMI-1640, aminopterin, ovalbumin (OVA), hypoxanthine, thymidine, goat anti-mouse immunoglobulin G-horseradish peroxidase conjugate (IgG-HRP), 3,3′,5,5′-tetramethylbenzidine(TMB),1-Ethyl-3-(3-dimethylaminopropyl)carbodiimide hydrochloride (EDC·HCl), *N*-hydroxysuccinimide (NHS), and Freund’s adjuvants (complete and incomplete) were all obtained from Sigma–Aldrich (St. Louis, MO, USA). Other biochemical reagents were obtained from Roche (Basel, Switzerland), including, l-Gla (l-Glutamine), and HFCS (Hybridoma Fusion and Cloning Supplement). The Hainan Province Centres for Disease Control and Prevention supplied the Pasteurella negative female BALB/c mice. The China Centre provided the SP2/0 myeloma cells for Type Culture Collection (CCTCC). The culture flasks and cell culture plates (6, 24, and 96 wells) were purchased from Costar (Corning, NY, USA).

The TLC tests were performed while using ultraviolet (UV) light and sprayed with 5% H2SO4 in C2H5OH (*v/v*) for observation. The 1H-NMR spectrum was carried out on a Bruker AM-400BB instrument (Bruker, Karlsryhe, Germany) with TMS serving as an internal standard, operating at 400 M Hz. A microplate strip washer (Elx50, Bio-Tek Corp, Winooski, VT, USA) and a microplate reader (Elx800, Bio-tek Inc, Winooski, VT, USA) was used to perform the ELISA method. The HPLC (Ultimate3000, Thermo Fisher Inc., Waltham, MA, USA) detection was used to validate the developed assay in the spiked samples. Chem Office (Chem 3D and Chem Draw, 15.0 version, Bedford, MA, USA) was used for drawing the planar and spatial structure of the compounds.

### 4.2. Synthesis of Hapten

According to a previous report [19], the hapten that is shown in Figure 4 was prepared by reducing the nitro group to the amino group, and linking it with the succinic acid while using the following operations: 27.9 g of SnCl_2_·2H_2_O was added into the reactor containing 5 g of dinotefuran dissolved in ethanol. The mixture was stirred and heated at reflux under the protection of N_2_ for 4 h, and one outlet of the reactor was subsequently opened to allow for the evaporation of ethanol. Sufficient cold water was added to the reactor with violent shaking until no more precipitation was generated. Finally, ethyl acetate was added to isolate the product (faint yellow oil) six times, and the product was reacted with 1.2 times of succinic anhydride that was dissolved in dichloromethane at room temperature for 8 h. The mixture was purified two times to exclude the remaining succinic anhydride. The final product was collected and characterised while using NMR.

### 4.3. Preparation of Antigen (Hapten-Protein Conjugates)

The immunogen (antigen in Figure 4) and the coating antigen (hapten-OVA conjugate) were obtained by coupling the hapten (**4**) with BSA and OVA, respectively. Both immunogen and the coating antigen were prepared by the diazotization method [20,21]. EDC·HCl (0.04 g) was added to the PBS buffer (0.01 mol/L) that contained 0.1 g BSA and stirred for 15 min. to dissolve the mixture. 0.02 g of NHS was added to the solution and stirred for 40 min., and then the pH value of the solution was adjusted to 8.0. Next, 1 mL DMF containing 0.03 g of hapten was added to the solution drop-wise and then stirred overnight to prepare the immunogen. The coating antigen was synthesized with the same method. Based on the immune dose, the immunogen and coating antigen were cryopreserved after split charging to avoid repeated freeze-thawing cycles.

### 4.4. Immunization

BALB/c female mice that were aged 6–10 weeks were subcutaneously immunized with the immunogen (150 μg) in physiological saline solution and complete Freund’s adjuvant (*v/v* =1:1) for the initial injection [22]. Additionally, other three immunizations were intradermally given at the same dose, with incomplete Freund’s adjuvant and physiological saline solution (*v/v* =1:1) after the initial injection on day 21, 35, and 50, respectively.

The titer was prepared seven days after the last injection, and used to determine the specificity of the antibody in the antiserum obtained from tail blood using indirect ELISA and competitive indirect ELISA (CI-ELISA). The final intraperitoneal injection was given to the mice showing high response three days before cell fusion, and its spleen was used for the cell fusion procedure.

### 4.5. Cell Fusion and Cloning

The splenocytes were fused with the sp2/0 myeloma cells at a ratio of 1:10 by a chemical method in the presence of polyethylene glycol (PEG, molecular weight 1450) [23]. The hybridoma cells were cultured in the semisolid complete medium hypoxanthine containing methylcellulose, FBS, thymidine, aminopterin, and HFCS, and equal volumes of the cell suspension were cultured into the 96-well plates [24]. Approximately 14 days later, the supernatant from each well was subjected to CI-ELISA; supernatants with an absorption value >3.5 were transferred to the 24-well microculture plates with a complete RPMI-1640 medium containing thymidine, hypoxanthine, and 20% FBS.

The cells with no cross-reactivity with OVA but good reactivity with dinotefuran were chosen for further analysis using indirect ELISA and CI-ELISA screening. Further subcloning experiments were performed in six-well culture plates while using the semisolid complete medium to ensure the mono-clonality of the cells. The clones, stably producing high-quality antibody, were passaged and cryopreserved in a freezing solution according to the following protocol: 40 min. at 4 °C, 1 h at −20 °C, and 5 h at −70 °C, followed by storage in liquid nitrogen. Mice ascites was collected and purified to obtain the McAbs [25].

### 4.6. Non-Competitive ELISA and Competitive Indirect ELISA

The procedures of the CI-ELISA assay were similar to those that were described in a previous study [26]. Briefly, the plate was washed with PBST (Tween in PBS, 1.14 g/L Na_2_HPO_4_, 20.7 g/L NaCl, 0.19 g/L KCl, 0.19 g/L KH_2_PO_4_, and 0.05% Tween, *v/v*) after coating overnight at 4 °C with 100 μL/well of the coating antigen dissolved in the coating buffer (2.92 g/L Na_2_CO_3_ and 1.58 g/L NaHCO_3_).

The wells were blocked with OVA solution (1% in PBST, 200 μL/well) and then incubated at 37 °C for 2 h. Thereafter, analytes (50 μL/well) and 50 μL/well of antiserum diluted with PBST (or only 100 μL/well of antiserum for determination of antibody titer) were added to the wells and incubated at 37 °C for 1 h. The anti-mouse-IgG-HRP conjugate (100 μL/well) was added to specifically bind to the antibody on the plates at 37 °C for 1 h. Finally, the enzyme-catalysed reaction was stopped by the addition of 50 μL/well of stop solution (2 M H_2_SO_4_) after adding the substrate solution (containing 9.5 mL of PBS buffer (pH = 5.0), 0.5 mL of 2 mg·mL^−1^ TMB (dissolved in ethanol), and 32 μL of 3% (*w/v*) urea hydrogen peroxide) at 37 °C for 15 min. The absorbance of the mixture was immediately measured at the dual-wavelength mode at 450 nm. The analyte was added to compete with the coated antigen, and this step was the only difference between the CI-ELISA and the non-competitive ELISA.

### 4.7. Affinity, Sensitivity, and Specificity

In the non-competitive ELISA, the titer of the antibody was determined by measuring the affinity of the antibody to the antigen by assessing different serial combinations of the antibodies and the corresponding coating antigen. The sensitivity of the antibody, being expressed as the IC_50_ value (the concentration at which the binding of the antibody to the coating antigen is inhibited by 50%), was calculated from the curve that was obtained from the gradient concentrations of analytes in the CI-ELISA procedure. The cross-reactivity (CR), which is an indicator of the antibody specificity, was obtained according to the ratio of the IC_50_ of dinotefuran (assigned as 100%) to that of the test compounds, such as acetamiprid, dinotefuran, nitenpyram, diuron, and tetrahydrofuran.

### 4.8. Optimization of the CI-ELISA

Several experimental parameters (organic solvent concentration and buffer) were studied to obtain the best performance. Among them, we studied the lowest IC_50_, which was the primary criterion for evaluating the immunoassay. The effects of the organic solvent (10, 20, 40, and 60% of methanol, *v/v*) and the reaction buffer (PBS and PBST) on the performance of the immunoassay were studied while using the optimized working concentrations of the antibody and antigen. A quantitative standard curve of the precision versus sensitivity was plotted from a repeated CI-ELISA assay (n = 3) while using the optimized parameters.

### 4.9. Analysis of Spiked Samples

A spiked recovery experiment was performed to evaluate the accuracy of the developed assay method. Cabbages that were obtained from a local supermarket were spiked with three concentrations of dinotefuran (1, 5, 10 ng/g) and pre-treated using the QUENCHERS method (17, 18). They were then analyzed while using the developed ci-ELISA and HPLC method. The LC method was performed at 30 °C using isocratic elution (methanol and water, *v/v* = 2:8) and C_18_ column (Waters, Milford, MA, USA). The signal was detected at 270 nm.

## Figures and Tables

**Figure 1 molecules-24-02426-f001:**
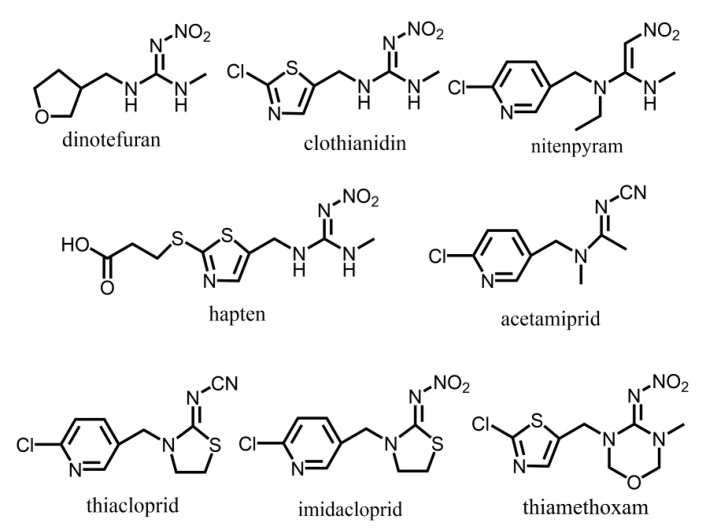
Molecular structures of analytes and hapten.

**Figure 2 molecules-24-02426-f002:**
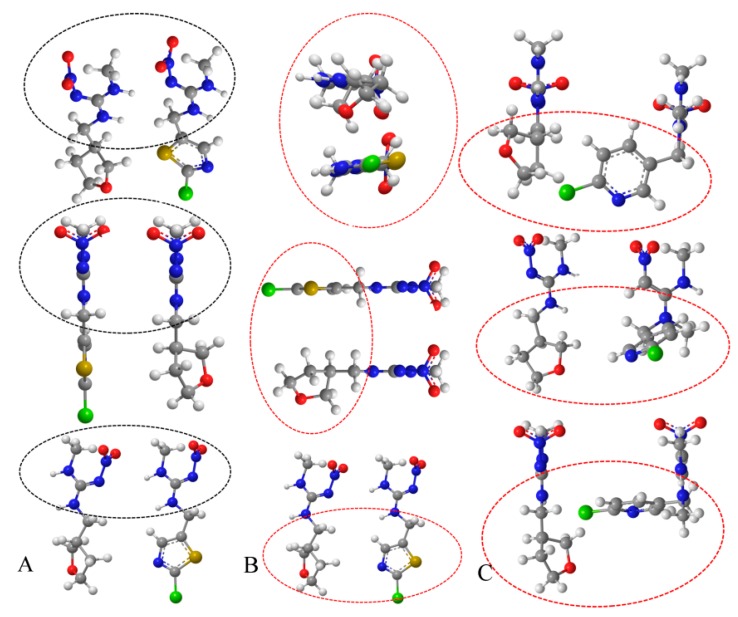
Spatial structures of analytes (The analyte with a green atom was clothianidin in column **A** and **B**, and correspondingly, it was nitenpyram in column **C**. The other molecular in each column is dinotefuran. The black and red circle in the figure manifested the same and different structure of the analytes, respectively.).

**Figure 3 molecules-24-02426-f003:**
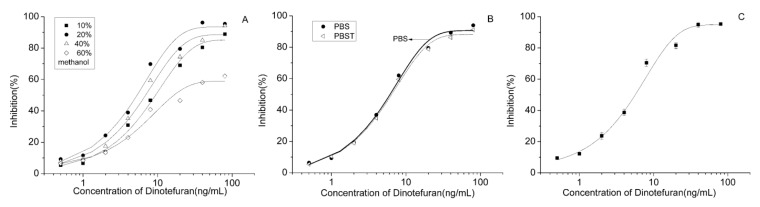
Inhibition curves of the antibody (**A**), optimization to the methanol proportion; (**B**) selection between PBS and PBST; and, (**C**) the optimized curve).

**Figure 4 molecules-24-02426-f004:**
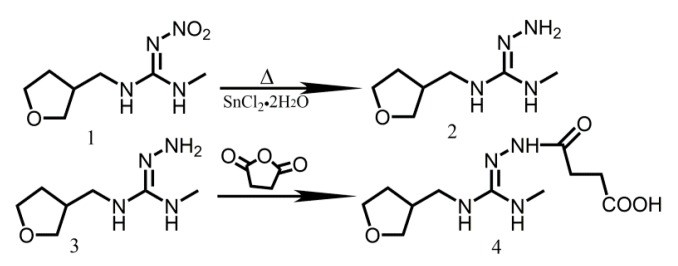
Synthesis route of antigen.

**Table 1 molecules-24-02426-t001:** Titre and IC_50_ of the antibodies.

Antibody	Titer	IC_50_ (mg/L)
2D6	160,000	29.0 ± 0.23
2F10	128,000	24.1 ± 0.12
3C7	64,000	6.17 ± 0.05

**Table 2 molecules-24-02426-t002:** Cross-reactivities (CRs) of the antibody to the analytes.

Analyte	Structure	CR(%)
2D6	2F10	3C7
Dinotefuran	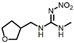	100
Acetamiprid	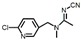	4.90	4.37	1.44
Nitenpyram	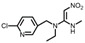	5.67	5.06	1.13
Diuron	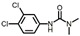	0.17	0.20	0.14
Thiamethoxam	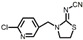	<0.096	<0.096	<0.048
Clothianidin	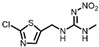	0.11	0.20	<0.096
Tetrahydrofuran	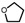	9.1	6.7	6.1

**Table 3 molecules-24-02426-t003:** Recoveries of the dinotefuran in enzyme-linked immune assay (ELISA) and liquid chromatography (LC) methods.

Spiked Concentration (ng/mL)	Average Recovery (n = 3)		RSD (%)	
	ELISA	LC	ELISA	LC
1	91.02	90.06	4.31	1.59
5	98.45	92.76	4.08	1.88
10	87.49	96.96	4.75	4.43

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
