# Peer review of "Development of a High-Quality ELISA Method for Dinotefuran Based on a Novel and Newly-Designed Antigen"

_molecules, 2019, doi:10.3390/molecules24132426_

Round 1

Reviewer 1 Report

The work is technically good but it suffers from some weaknesses that lead me to not recommend publication in its current form. Some particular comments are:

1-    Can the authors comment on “the lowest IC50 (5.83 ng/mL) was obtained at 20% methanol”?

2-    The authors use 20% methanol and they know that inhibits the in somehow the Ab, why not trying to optimize lower values?

3-    On line 185 the authors claim, “although they were slightly lower than those of the HPLC method”, you mean its higher??? Not lower!!

4-    Can the authors comment on “ELISA method designed can be further modified through field testing and used as a rapid detection technique”

Author Response

To reviewer 1

Thanks a lot for your helpful comments and suggestions, and we addressed your comments point by point.

Reviewer’s comments:

The work is technically good but it suffers from some weaknesses that lead me to not recommend publication in its current form. Some particular comments are:

1-    Can the authors comment on “the lowest IC50 (5.83 ng/mL) was obtained at 20% methanol”?

This procedure in the work is aim to find an appropriate methanol proportion for dissolving the analyte. It is not the final optimization results. The results showed the lowest IC50 (5.83 ng/mL) was obtained when 20% methanol was used for resolving the analyte. Methanol proportion is an important factor that affects the sensitivity of the antibody. Low concentrations of methanol affected the solubility of the analyte, whereas high proportions of methanol reduced the activity of the antibody. This value is only the result obtained in the optimization to methanol proportion, which is a relatively excellent result of antibody sensitivity. Probably, it is not the highest sensitivity performance of the antibody because not all of the conditions were optimized.

2-    The authors use 20% methanol and they know that inhibits the in somehow the Ab, why not trying to optimize lower values?

10% methanol was already used for the optimization but no better result was obtained, line 182, figure 3.

3-    On line 185 the authors claim, “although they were slightly lower than those of the HPLC method”, you mean its higher??? Not lower!!

This is a clerical error from the authors. Here, authors wanted to illustrate the shortcomings of the developed method compared with the HPLC method. Therefore, ‘they’ in the sentences should be the RSD of the recoveries. The whole sentence had been revised, line 204-205 in the revised manuscript.

4-    Can the authors comment on “ELISA method designed can be further modified through field testing and used as a rapid detection technique”

Antibody and ELISA is the basis of some rapid detections, such as test strip and fluoroimmunoassay detection. “through field testing” is not appropriate and the sentence should be revised as follows, “ELISA method designed can be further modified and used for a rapid detection technique in field testing.”, line 208-209.

Reviewer 2 Report

The manuscript describes the synthesis and characterization of an antigen mimic for the generation of antibodies. Well done.

Author Response

To reviewer 2

Thanks a lot for your helpful comments and suggestions, and we addressed your comments point by point.

Reviewer’s comments:

The manuscript describes the synthesis and characterization of an antigen mimic for the generation of antibodies. Well done.

The query that “Does the introduction provide sufficient background and include all relevant references?” was also improved, line 60-68.

Reviewer 3 Report

This manuscript presents a new method for the determination of dinotefuran in spiked cabbage samples. Authors state that they new design of the hapten contributes to improve the cross-reactivity to clothianidin of conventional ELISA assays. However, many aspects remain unclear, such as those described below:

1. Authors state that they can design the tracer based on spatial simulation, but they do not mention the software they have used.

2. Chemical structures of the compounds provided as 3D views are difficult to see and understand the reasons provided by the authors.

3. Authors do not check the structure of tetrahydrofuran with the software. This means that there is not possibility for spatial simulation with this compounds, which presents the highest cross reactivity. Authors do not explain why the method is somehow sensitive to tetrahydrofuran with a low cross-reactivity percentage

4. Authors should have checked the performance of  the proposed ELISA method with the commercial kits in order to demonstrate the feasibility of both for the analysis of cabbage samples.

For this reason, this reviewer considers that the manuscript cannot be accepted in its present form.

Author Response

To reviewer 3

Thanks a lot for your helpful comments and suggestions, and we addressed your comments point by point.

Reviewer’s comments:

This manuscript presents a new method for the determination of dinotefuran in spiked cabbage samples. Authors state that they new design of the hapten contributes to improve the cross-reactivity to clothianidin of conventional ELISA assays. However, many aspects remain unclear, such as those described below:

1. Authors state that they can design the tracer based on spatial simulation, but they do not mention the software they have used.

The software used in this work was Chem Office (15.0 version), and its information had been added in the text, line 258-259.

2. Chemical structures of the compounds provided as 3D views are difficult to see and understand the reasons provided by the authors.

Spatial structure of hapten is a vital factor that affects the reaction between antibody and antigen. Thus, to know the spatial structure of hapten and analyte is an effective way to design an excellent hapten. 3D view of the compounds is helpful to understand the combination relationship between antibodies and antigens. Spatial structure views of the compounds in different directions were supplied in figure 2, and similarities and differences between haptens and analtes would be spatially exhibited in a same direction. It is evidently exhibited and marked in figure 2. It is helpful to know how the spatial structure of antigen affected the characteristic of the antibody, which also supplied evidence for further design of hapten. Thus, authors believed that figure 2 is necessary and helpful to understand the perspective about antigen design of the manuscript.

3. Authors do not check the structure of tetrahydrofuran with the software. This means that there is not possibility for spatial simulation with this compounds, which presents the highest cross reactivity. Authors do not explain why the method is somehow sensitive to tetrahydrofuran with a low cross-reactivity percentage

Tetrahydrofuran’s structure is the same with the furan structure in dinotefuran, and it’s spatial structure is also the same. Thus, there is no necessity to compare its structure with others. At the same time, the results also showed that the antibody showed the highest cross-reactivity to tetrahydrofuran. Small and highly similar molecular of tetrahydrofuran makes it easily combine and react with antibody, thus, high cross-reactivity was obtained. Relative speculation and explanation was made in the revised manuscript, line 152-156.

4. Authors should have checked the performance of the proposed ELISA method with the commercial kits in order to demonstrate the feasibility of both for the analysis of cabbage samples.

We totally agree with the reviewer that commercial kit is feasible and also further development of ELISA or antibody. However, HPLC or GC method is more often in detection standards in many countries and areas, actually, equipment methods are also more stable and widely accepted although they are complex, expensive and time-consuming. Commercial kits are also usually checked with HPLC or GC methods. Thus, checking the performance of the ELISA method with HPLC method is more credible.

The queries

Does the introduction provide sufficient background and include all relevant references?”

The introduction was improved, line 60-68.

Is the research design appropriate?

Furan and guanidine are two parts of molecular structure of dinotefuran, and the hapten with guanidine as the characteristic was also reported. In this work, new hapten was designed compared with the reports. Accordingly, the results in different works showed that antibodies are closely related with spatial structure of hapten. Finally, spatial structures of analytes were compared and analyzed considering the quality of antibodies. The route design of hapten is clear and the evidence of result is effective. Relationship between spatial structure of hapten and quality of antibody is directly proved.

Are the conclusions supported by the results?

The conclusion was revised to develop close relationship with the results, line 218-230.
